# Network evolution of diffusion in enterprise digitalization and intellectualization transformation: A technology—organization —environment framework perspective

Huan Hu ⊙*, Mingyu Zhao, Xiaoyi Zhang

School of Economics & Management, Northwest University, Xi'an, China

* hu_huan1121@163.com

## Abstract

Enterprise digitalization and intellectualization (EDI) is a crucial aspect of China's modernization process. However, uncertainty in market and business decisions hinders the EDI diffusion process in China. Therefore, this research aims to solve the uncertainty problem of EDI diffusion by examining market demand and government policy coordination. First, we utilize complex network game theory and establish a technology–organization–environment framework for the factors that influence the elements of EDI transformation by combining evidence from existing studies. Second, the network game model is constructed to analyze and optimize the updating rules in the network as a diffusion strategy that enterprises under the uncertain market can adopt. Finally, the impact of adjusting government subsidies and different premiums on the diffusion of EDI transformation strategies is examined. The degree of market diffusion and average revenue of EDI are higher after the optimization of network node strategy updating rules compared to before optimization. Further analysis reveals that only the premium effect of product pricing and inverted U-shaped subsidy support from the government affect the degree of market diffusion and the average revenue of EDI, while the other premium effects are not significant. These findings enrich research related to complex networks and nonlinear dynamic strategies. They also indicate recommendations for government policies to enhance diffusion efficiency and reasonable pricing for enterprises to promote returns.

## Introduction

The 20th Party Congress highlighted the importance of accelerating the integration of industrial digitalization and intellectualization (DI) to promote high-quality economic development. DI plays a crucial role in achieving this objective. Currently, companies primarily use digital technologies to collect user information for product and service updates [1], resulting in data stock growth. However, the value of massive data has not been fully explored. According to data released by experts at the "Big Data and Artificial Intelligence" sub-forum of the 5th

**Data Availability Statement:** All relevant data are within the paper and its Supporting Information files.

**Funding:** Project of Humanities and Social Sciences Foundation of the Ministry of Education (Specific grant numbers: 18XJAZH004); Initials of authors who received each award: H H; Full names of commercial companies that funded the study or authors: Ministry of Education of China; Initials of authors who received salary or other funding from commercial companies: W Z; URLs to sponsors' websites: https://www.sinoss.net/. The funders had no role in study design, data collection and analysis, decision to publish, or preparation of the manuscript.

**Competing interests:** The authors have declared that no competing interests exist.

China Cloud Computing Conference, 99.6% of the data value in China remains untapped. Data captured by digital technology needs to be further explored for its value through the application of digital intelligence technology, thus, there is an urgent need to develop concepts other than digitization. DI is a "new concept" that combines digitalization and intellectualization to provide more accurate, innovative, and demand-driven products and services. It combines digital technologies such as blockchain, internet of things, and cloud computing with intelligent technologies such as artificial intelligence (AI) to improve operational efficiency. For instance, Haier established the Khaos industrial Internet platform after promoting DI, which has enabled users' full process participation, reduced inventory, improved efficiency, and facilitated the development of multiple subjects collaboratively [2]. Therefore, it can be inferred that DI, as the next phase of digitalization transformation, may emerge as a prominent trend.

However, there are fewer existing academic studies on DI, which mainly focus on education [3], library intelligence [4], and supply chain [5,6] studies. Although these studies have recognized the significance of DI transformation, they tend to take a macro perspective, overlooking the crucial role of enterprises as micro actors. DI transformation should be seen as an enterprise behavior that can drive technological innovation, promote cooperation between industry, academia, and research, and foster a virtuous cycle of enterprise digitalization and intellectualization (EDI) transformation. However, existing research on EDI is limited: it mainly focuses on enterprise digitalization or intelligent transformation [7–9]. To some extent, these studies provide a foundation for the study of enterprise digital or intelligent transformation but hinder our understanding of EDI behavior from a mixed perspective. Furthermore, scholars have considered combining digitalization and intelligence to study the transformation process of enterprises [10].

Existing research methods on technology transformation such as digitalization or intelligence have taken several different approaches. For enterprise digitalization or intelligence, Han et al. established a digital maturity model to evaluate the degree of digital transformation of consulting firms and showed that future digital transformation requires comprehensive integration of more specialized digital technologies [11]. Although some scholars have adopted an econometric approach to study the impact of single (multiple) factors on technological transformation such as digitalization or intelligence of enterprises, such as R&D [12], policy [13], and resources [14], it does not view enterprise transformation as a systemic change. A game theory approach can further study the interactions between stakeholders and the impact of their own decisions on the diffusion of technological transformation, such as digitalization or intelligence. Wang et al. used a three-party game model to study the impact of heterogeneous governmental environmental regulations on the diffusion of green technology in the manufacturing industry from the perspective of supply and demand [15]. Zhou et al. established a non-cooperative game model to study ways to balance the competitive interests of customers and reduce the cost of manufacturing technology in the process of the transformation of shop-floor intelligence in an enterprise [16]. Yin et al. used a stochastic differential game model to consider the diffusion of low-carbon technology sharing in a collaborative innovation system between advantaged and disadvantaged firms considering internal and external influences [17].

EDI can be viewed as a game evolution process among stakeholders in the market, such as enterprises, government, and the public. Xu et al. [18], Zhang et al. [19], and Brunswicker et al. [10] examined the role of different actors in promoting DI and EDI. However, most researchers focus on the interactions of different subjects in the system and seldom consider the interactions of the same kind of units within different game subjects. Meanwhile, Yin et al. attempted to use complex networks to describe the sustained epidemic prevention effect of

government collaboration under the Big Data Intelligence Innovation (BDII) mechanism [20], while Tan et al. further explored the evolution of the information sharing behavior of the network of firms within the supply chain by using the game methodology on the complex network, discussing the information synergistic effect of the firms in the supply chain and the strategies that can be adopted in the process of the diffusion of behavior [21]. In addition, most game models use static parameter mechanisms for simulation [22], and these models may not be able to comprehensively capture dynamic changes in strategies during the evolution of the game [23].

Building on the preceding analysis shows that the model of a complex network game can be used to solve the problem of digital and intelligent transformation of enterprises. However, most existing research examines the transformation process of enterprise digitalization or intelligence separately, or conflates the two concepts, and few studies on digital and intelligent transformation exist. The research methods mostly use a single two- or three-party game form for digital or intelligent research, without considering that multiple enterprises are affected by the decisions of neighboring enterprises in the DI diffusion process. Moreover, the fixed parameters also allow participants access to only a static and single management mechanism. Therefore, this study aims to examine the factors influencing the transformation of EDI using a technology–organization–environment (TOE) framework of technology innovation diffusion. Specifically, the study develops a complex network game model of EDI transformation diffusion, and design static, linear dynamic, and non-linear dynamic subsidy and premium schemes. Simultaneously, we optimize the update rules, which can guide enterprise choosing strategies under the uncertain market(complex network), and analyze the effectiveness of three different subsidy and premium schemes, corresponding to static, dynamic linear, and dynamic inverted-U types, in combination with simulations.

Given that uncertainty in government and business decisions hinders EDI transformation, research on the evolutionary process of EDI diffusion and the strategies adopted by stakeholders in market environments is necessary. Therefore, this research aims to address the uncertainty of EDI diffusion by examining market demand and government policy coordination. Regarding the existing background, this study has the following significance: theoretically, this paper explores digital intelligence that is different from digitalization; establishes a complex network game model; optimizes the updating rules of the network game as a corporate strategy for EDI diffusion; further expands existing complex network game theory; enriches the application of TOE theory in the transformation of digital intelligence; and identifies the relevant technological, organizational, and environment-related influencing factors. At the same time, the advantages of nonlinear parameters in complex network diffusion are verified; practically, the updated diffusion rules provide a better strategy for enterprises to carry out DI diffusion. Furthermore, the setting of nonlinear parameters in the network game provides a management basis for government and enterprises in the process of DI transformation. The findings of the study can provide valuable insights into factors that drive EDI transformation.

The paper is structured as follows: The *Literature review* section describes and constructs the research framework of the diffusion of EDI transformation based on TOE theory. The *Research hypotheses* section presents the research hypothesis. The *A network game analysis of diffusion of EDI* section presents the network game model and network update rule. The *Simulation analysis and discussion* section describes the process of simulation analysis and discussion, including dynamic subsidy and premium effect. The *Conclusions and implications* section contains the conclusion and implications.

## Literature review

### Technology diffusion and complex network games

Technology diffusion research is an effective means of breaking the shackles of digitalization in existing enterprises. As DI is a form of technology integration diffusion, it shares similarities with the diffusion of other technologies. Research on technology diffusion is extensive; Guo et al. employed the Lotka-Volterra model to examine the diffusion of building information modeling (BIM) technology operation and maintenance, and identified three diffusion results: full market share, partial market share, and exit from the market [24]. Similarly, Benhabib et al. investigated how endogenous innovation and technology diffusion interact to shape productivity distribution and generate aggregate growth. They discovered that expensive technological inputs create a gap between the best and worst technologies used to produce similar goods, and that the aggregate growth rate equals the innovator's maximum growth rate in the equilibrium growth path [25]. While the diffusion results of EDI transformation are likely to share some characteristics with technology diffusion, DI transformation integrates and applies DI technologies. Therefore, diffusion of DI technology should be distinguished from diffusion of a single technology. This requires considering not only the combined cost of different technology applications but also the important roles of government and the public in promoting the DI process. Moreover, companies in the market are influenced by themselves as well as by the mutual influence of different companies in the same market.

The study of games in complex networks began with the prisoner's dilemma game in the rule network, which compared individual and neighbor gains and adjusted game strategy by imitating the optimal rule. The research found that cooperative participants could counteract the betrayal strategy by aggregating on the rule grid [26]. Since then, new developments in games in complex networks have emerged. Chen et al. investigated prisoner's dilemma games in scale-free community networks and found that a more pronounced community structure leads to higher levels of group cooperation [27]. Liu et al. addressed the problem of individual diversity and independent networks in the evolution of cooperative behavior, transforming the game model into an interdependent network by hybrid coupling (i.e., of utility and probability). They classified players' strategic behavior types into a layer related to individual diversity and found a hybrid coupling optimal region between networks that can promote cooperation [28]. Studies have also examined innovation diffusion in the market for a particular technology or product based on a complex network game. Wang et al. explored the competition mechanism between firms and neighboring networks in the industry using a complex network evolutionary game model [29]. Zhao et al. modeled diffusion of new energy vehicles based on different networks and found that macro-level government subsidy rates can promote the diffusion of new energy vehicles, while large networks are more sensitive to subsidies than small networks [30]. Therefore, combining complex networks with analysis of EDI transformation in the market can more clearly identify the relevant mechanisms in the diffusion process.

### An EDI transformation framework

The term "digital intelligence" was initially introduced by the "Knowledge-Based Consortium" at Peking University, which defined it as a combination of digital intelligence and intelligent digitalization. Scholars have defined DI as the integration of digital and intelligent technology [31]. As a crucial aspect of China's DI transformation, the DI transformation of enterprises can be defined as the interactive application of DI technologies to enhance the profitability and efficiency of enterprises. This can also be considered as the proliferation of multiple technology combinations in various enterprises within the market [32].

Extensive research has investigated technology diffusion, and found that TOE theory is a well-established framework that explains the emergence and development of new technologies in the market. The TOE framework was proposed by Nazky and Fleischer in 1990 as a comprehensive tool for analyzing the impact of factors such as technology, organization, and environment on new technology adoption. The framework's strengths are its strong systematicity, flexibility, and operability. As the framework does not specify explanatory variables for the three types of factors, it remains widely applicable in various research scenarios and purposes. Hence, the TOE framework has become an indispensable analytical tool in scientific research for studying the factors influencing new technology adoption [33]. TOE theory has been widely used in the study of supply chains [34,35], BIM technology applications [36,37], and innovation patent development [38,39].

Based on the above research, the market diffusion path of EDI transformation needs to be considered from three levels. First, at the technical level, technology R&D investment will enhance the degree of specialization of enterprises for the application of DI technology [40]. Simultaneously, the research and development of new technologies will also enable the enterprise to enhance the corresponding knowledge base, and accumulate knowledge of DI transformation [41]. The technology premium effect has a positive impact on the economic consequences of product returns, production, and operational efficiency [42], and the economic foundation determines the superstructure, which can increase investment in the research and development of digital intelligence technology and the introduction of relevant talent with the support of a sufficient economy. Second, at the organizational level, human resources play a crucial role in promoting EDI transformation. Human resources are an important cost for enterprises, which can not only promote the transformation of knowledge results generated by R&D investment, but may also reduce labor costs via the introduction of DI technology [43]. Third, at the environmental level, enterprises are not only in a competitive relationship with similar firms but also affected by heterogeneous consumers in the marketplace and government policies. Consumers' consumption concepts advance the important driving force of EDI transformation [44], which can source operating funds as well as enhance the enterprise's reputation and promote the sustainable development of the enterprise. Simultaneously, only by adhering to government policies can we better seize opportunities in the international environment, realize the social value of enterprises, and meet the needs of national economic development [45]. Existing studies identify different relevant influencing factors related to these three aspects, summarized in Table 1.

In summary, based on relevant research on the influencing factors of DI transformation, important factors at different levels of the TOE framework were selected as the follow-up research objects. The specific research factors include R&D investment (R&D) and technology premiums (digital technology and IT support) at the technical level, employee costs (human capital), and production and operation level (financial resources and business level) at the organizational level, and enterprise competition and cooperation (sectorial relevance), consumer preference (consumer preference), and government policy (government support) at the environmental level. Interaction among these factors can further enhance the DI level of the enterprise. Technological progress has resulted in automatic accumulation and precipitation of data flow, which not only improves the technical level but also enhances the management efficiency of the organization and the position of enterprises in the industry environment. Contrarily, organizational transformation can stimulate technological innovation and strengthen collaborative ability with other enterprises. Interactions at the environmental level can improve the efficiency of the entire society and promote further transformation of enterprise technology and organizational management, laying a solid foundation for EDI transformation. The government's policy support services also play a vital role in this transformation.

**Table 1. Factors influencing the transformation of DI technology at different levels.**

| Category | | Factor | Description | Studies |
|---|---|---|---|---|
| **EDI** | Technology *T* | Digital technology | Digital technology can help enterprises achieve digital management, enabling faster, more accurate, and automated production and business processes, promoting enterprise innovation, reducing costs, and improving profit margins. | Alexopoulos et al. [46] Trung et al. [40] Appio et al. [47] Tsiavos and Kitsios [41] Al-Ruithe et al. [48] |
| | | AI technology | AI technology can flexibly meet consumer needs, further expand service channels, reduce energy resource consumption in the supply chain, minimize production costs in various links, promote continuous improvement of production efficiency and profit margin, reduce the occurrence of human errors in the production process, and improve product quality. | Yeh et al. [49] Turel et al. [50] Weill et al. [51] Verhoef et al. [52] |
| | | R&D | R&D resources refer to the human, material, and financial resources necessary for innovation activities. | Ancillo and Gavrila [12] Zapata et al. [53] |
| | | Premium effect | Technology can generate product returns, production and operational benefits, and other benefits that exceed the value of technology itself. | Miyao [42] |
| | Organization *O* | Financial resources | The cost of implementing and supporting the project, including procurement, delivery, hardware and software installation, staff, and management training costs. | Omrani et al. [54] Kutnjak [55] Jones et al. [14] |
| | | Business level | Used to measure a company's business level and development stage, which can include enterprise size, skills of management and leadership. | Benlian and Haffke [56] Hansen et al. [57] Singh and Hess [58] Jackson and Dunn-Jensen [44] |
| | | Human capital | Managers who are proficient in the basic properties of digital technology and developing skills for existing workers and future digital workforce. | Nadkarni and Prügl [59] Holmstro and Nyle [60] Al-Alawi et al. [61] Hess et al. [62] Colbert et al. [63] |
| | Environment *E* | Sectorial relevance | The relationship between industry characteristics and competition, collaboration, and imitation between enterprises. | Martínez-Caro et al. [64] Kane et al. [65] |
| | | Government support | Introduce policies, mainly including subsidies and tax exemptions. | Wang et al. [45] Manda and Backhouse [13] Zhang et al. [66] |
| | | Consumer preference | Consumer preference is a personalized preference that reflects the degree of consumer preference for different products and services, and is an important factor affecting market demand. It is primarily determined by the impact of local social environment, customs, fashion changes and so on, on the entire consumer group or a specific group at that time. | Dehnert et al. [44] |

Thus, the TOE framework can provide a theoretical basis for constructing and verifying the complex network game model for the transformation of EDI under different scenarios of subsidy and premium schemes, as shown in Fig 1.

## Research hypotheses

In a perfectly competitive market, firms aim to maximize profits by selecting a production technology that aligns with consumer preferences and generates demand for their product. With the advent of the digital era, digital and intellectual enterprise has emerged as a leading market force with significant potential for growth and revenue. Within a DI diffusion network, firms continually update their connections and engage in imitation behavior, resulting in a homogeneous group effect that follows a power-law distribution of connected nodes. This

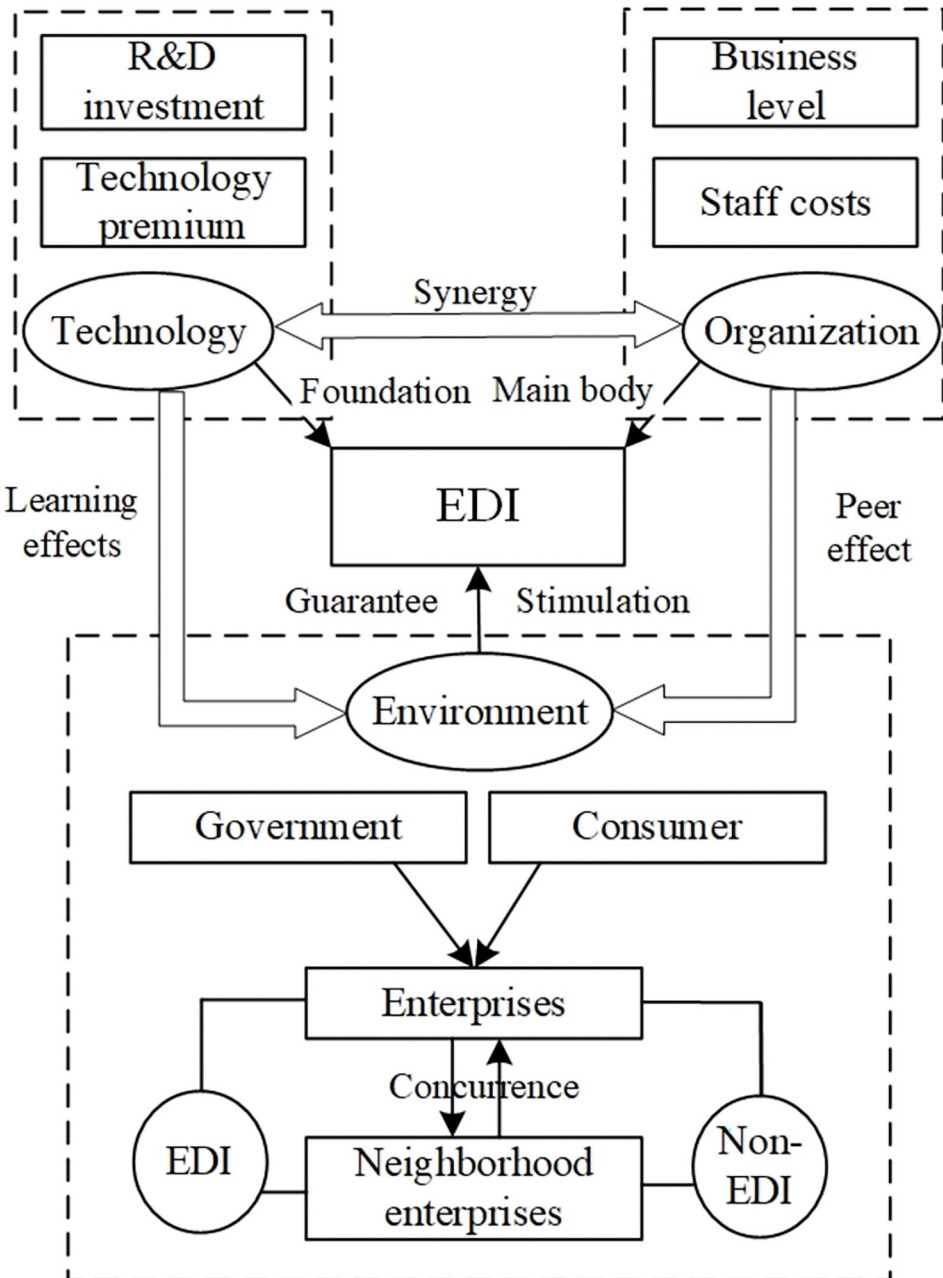

**Fig 1. TOE framework for the transformation conditions of EDI.** *Note*: R&D, research and development; EDI, enterprise digitalization and intellectualization; DIE, digital and intellectual enterprise.

network characteristic is indicative of a small-world network structure [67]. Based on the aforementioned market dynamics, we make the following assumptions.

**Hypothesis 1:** In a market comprising $N$ firms, proportion $x$ chooses to undergo a transformation in the DI sector, while the remaining $1-x$ do not. In a scale-free network, firms initially make one of these two decisions based on market diffusion. When an enterprise opts not to undergo DI transformation, it adopts traditional digital means to generate product income ($R_a$), other income (such as production and operation efficiency improvement) ($P_a$), and labor costs ($C_a$). In contrast, if the enterprise chooses to promote DI transformation, it must first

invest in talent, technology, equipment, and related areas ($I$). Owing to the adoption of advanced DI means, the enterprise's product income increases to $R_i$, while other income becomes $P_i$, and labor cost becomes $C_i$. The premium effect of DI transformation on product revenue, other revenue, and labor cost is determined by premium factors (α1, α2, α3). Studies suggest that the relationship between these factors is such that $R_i = R_a(1+\alpha_1)$, $P_i = P_a(1+\alpha_2)$, and $C_i = C_a(1-\alpha_3)$.

**Hypothesis 2:** The proportion of consumers with more DI product preferences is $\beta$, and those with traditional product preferences is 1-$\beta$. Consumer preference theory posits that consumer behavior is determined by their preferences. When facing the choice between traditional and DI products in the market, the profits generated by different enterprise strategies can vary depending on consumers' preferences. For instance, in the preliminary stages of the launch of new energy vehicles, owing to immature technology, most consumers preferred traditional products, and traditional vehicles remained the main source of sales profits for enterprises. However, with the continuous development of DI technologies, such as intelligent driving, planning, key element perception, and energy saving, automobile consumers' preferences have shifted toward DI products. These consumers are willing to pay higher prices for products with more DI characteristics. This shift has resulted in a significant increase in the proportion of sales profits from new energy vehicles in enterprises, which has further incentivized the automobile industry to prefer the production of DI products. When firms cater to consumers with DI product preferences: (1) If both the firm and its neighbors promote DI transformation, the unit product revenue is $f_i$, and the firm satisfies the average market demand $q$, resulting in product revenue $R_i = f_i \cdot q$; (2) if a firm promotes DI transformation, but its neighbors do not, consumer demand is satisfied by all firms that adopt DI transformation strategies (total number $x \cdot n$), and the firm receives product gain $R_i = f_i \cdot q/x$. In this scenario, neighboring firms do not receive external gains, resulting in a total gain of 0. When different firms cater to consumers with traditional product preferences: (1) If neither the firm nor its neighbors promote DI transformation, the unit cost is $f_a$, and the firm satisfies the average market demand $q$, resulting in product gains $R_a = f_a \cdot q$; (2) if the firm does not promote DI transformation, but its neighbors do, consumer demand is satisfied by firms that do not adopt the DI transformation strategy (total number $x \cdot n$), and the firm obtains product gains $R_a = f_a \cdot q/(1-x)$. In this scenario, neighboring firms do not receive external gains, resulting in a total gain of -$I$.

**Hypothesis 3:** We posit that the government will actively promote EDI transformation and provide a subsidy of $b$ to each product with DI characteristics. If both the enterprise and its neighbors adopt the DI transformation strategy, the total product gain of the enterprise should be $R_i = (f_i + b\eta) \cdot q$, where $\eta$ is the subsidy coefficient regulated by the government. If the enterprise carries out the DI transformation strategy, but its neighbors do not, the enterprise and neighbors obtain product gains $R_i = (f_i + b\eta) \cdot q/x$. Based on the above assumptions, the important symbols and their meanings in the text are shown in Table 2.

## A network game analysis of diffusion of EDI transformation

According to the model assumptions, we will establish the complex network model of EDI transformation diffusion. Referring to Liu's research [30,68], the complex network game model of EDI transformation diffusion mainly comprises three parts: network structure, game model, and evolution rules. Among them, the constructed network structure can reflect the distribution of different strategies of homogeneous enterprises in the market over time; the game model can determine the situation in which all firms in the network keep changing with the strategy changes of neighboring firms at each point in time; and the network evolution

**Table 2. Symbols and their meanings.**

| Parameters | | | |
|---|---|---|---|
| $R_a$ | The main business benefits that enterprises can obtain through traditional digital means (mainly through products). | $R_i$ | The main business income obtained by the enterprise after choosing to undergo DI transformation. |
| $P_a$ | The other benefits obtained by enterprises through traditional digital means (except for the main business income, the inflow of economic benefits formed in daily activities such as selling goods, providing labor services, and transferring asset usage rights, which are not frequent, generally small in amount per transaction, and have a low proportion of income). | $P_i$ | Other benefits obtained by enterprises after choosing to undergo DI transformation. |
| $C_a$ | The labor cost consumed by enterprises through traditional digital means. | $C_i$ | The labor cost consumed by enterprises after choosing to undergo DI transformation. |
| $I$ | When enterprises choose to undergo DI transformation, they first need to invest in talent, technology, equipment, and other related investments. | $\alpha_1$ | The premium factor for EDI transformation product returns (referring to the production of DI products during the DI transformation process that can increase corresponding product returns). |
| $\alpha_2$ | The premium factor for other benefits of EDI transformation (referring to the use of DI technology and management methods in the digital transformation process that can enhance corresponding other benefits). | $\alpha_3$ | The premium factor for labor costs in EDI transformation (referring to the use of DI technology and management methods during the DI transformation process that can reduce corresponding labor costs). |
| $x$ | The proportion of enterprises that choose DI transformation, and the proportion of enterprises that do not undergo transformation is 1-$x$ | $\beta$ | The proportion of consumers who prefer products with more DI characteristics, The others who prefer traditional products is 1- β. |
| $f_i$ | Unit product revenue of enterprises and their neighbors undergoing DI transformation. | $f_a$ | Unit product revenue for enterprises and their neighbors that do not undergo DI transformation. |
| $\eta$ | Subsidy coefficient regulated by the government. | $n$ | The total number of enterprises in the market. |
| $q$ | Average market demand for products. | $b$ | On the basis of actively promoting EDI transformation, the government will provide subsidies for every product with DI characteristics. |

rule can determine the decision criteria of whether firms change according to the strategies of their neighbors. Thus, the study discusses the construction process of the model stepwise.

## Network game model under stochastic rules

**Network structure.** Research has demonstrated that small-world networks are a valuable tool for analyzing social networks that can be classified into two main types: NW small-world networks and WS small-world networks. NW small-world networks involve the random addition of edges, ultimately resulting in the connection of all nodes in the small world, which does not align with the reality that not all enterprises can influence each other. Conversely, the broken edge reconnection of WS small-world networks provides a more accurate representation of the changing nature of mutual influence among enterprises. In this study, the coupling of EDI technology innovation strategies on a small-world network is simulated and analyzed, using the WS small-world network algorithm [69]. Initially, a circular nearest-neighbor coupling network containing **N** nodes is established, with each node connected to **d** neighbors. Subsequently, in each period, nodes in the network will randomly disconnect an edge with probability **p**, and reconnect a new node, ensuring that the number of coupling points at each endpoint remains constant, according to the k-nearest neighbor algorithm in complex network theory.

**Game model.** The diffusion of DI technology within an enterprise refers to the gradual adoption of DI technology over time within the market of digitally qualified enterprises. The

**Table 3. Consumers have DI product preferences.**

| | | Neighborhood enterprise | |
|---|---|---|---|
| | | Adopt | Not Adopt |
| Enterprise | Adopt | $(f_i+b\eta)q+P_i-C_i-I$ <br> $(f_i+b\eta)q+P_i-C_i-I$ | $(f_i+b\eta)q/x+P_i-C_i-I$ <br> 0 |
| | Not adopt | 0 <br> $(f_i+b\eta)q+P_i-C_i-I$ | 0 <br> 0 |

extent to which DI innovative products and services resulting from the diffusion of DI technology are accepted within the entire market also plays a crucial role in the diffusion of DI technology. Thus, this study presents a game model for the diffusion of DI technology innovation, which is based on evolutionary game theory. The game payoff matrices under different DI transformation strategies of firms and neighboring firms under different consumer types can be derived based on the research hypotheses, as shown in Tables 3 and 4.

Drawing on the game payoff matrix presented in Table 1, the expected payoff of manufacturer $i$ adopting the DI strategy can be calculated based on several factors. Specifically, the number of neighbors that have adopted the DI transformation strategy among all the neighbors of firm $i$ is denoted as $n_i$, whereas the number of neighbors not adopting the DI transformation strategy is represented by $n_a$. Moreover, the proportion of consumers with DI characteristic product preferences is denoted as $\beta$. By taking these factors into account, the expected payoff of manufacturer $i$ adopting the $D_i$ strategy can be determined from Eq (1)

$$u_i = \{n_i \quad n_a\} \times \left\{ \begin{array}{cc} (f_i + b\eta)q + P_i - C_i - I & -I \\ (f_i + b\eta)q/x + P_i - C_i - I & -I \end{array} \right\} \times \left\{ \begin{array}{c} \beta \\ 1 - \beta \end{array} \right\} \tag{1}$$

Similarly, the expected return of adopting the $D_a$ strategy can be deduced from Eq (2)

$$u_a = \{n_i \quad n_a\} \times \left\{ \begin{array}{cc} 0 & f_aq/(1-x) + P_a - C_a \\ 0 & f_aq + P_a - C_a \end{array} \right\} \times \left\{ \begin{array}{c} \beta \\ 1 - \beta \end{array} \right\} \tag{2}$$

## Dynamic subsidies and premium effects

**Dynamic subsidies.** Several existing studies have proposed static subsidy schemes, where the parameter $\eta$ remains constant. In this study, the subsidy is represented as $b \cdot \eta$. However, scholars have suggested that static subsidies may not be sufficient and that governments should consider dynamic subsidy schemes that better reflect market conditions [70]. While traditional dynamic subsidy approaches can create a negative correlation between market share and subsidy, some scholars argue that subsidies should follow an alternative non-increasing functional form with a peak (such as an inverted U-shaped relationship) [71]. In this study, we set the positive correlation function as $\eta = -v\gamma+k$ and the inverted U-shaped relationship as $\eta = $

**Table 4. Consumers do not have DI product preferences.**

| | | Neighborhood enterprise | |
|---|---|---|---|
| | | Adopt | Not adopt |
| Enterprise | Adopt | -I <br> -I | -I <br> $f_aq+P_a-C_a$ |
| | Not adopt | $f_aq/(1-x)+P_a-C_a$ <br> -I | $f_aq+P_a-C_a$ <br> $f_aq+P_a-C_a$ |

$-u\gamma^2+i\gamma$. To ensure that the different subsidy forms have the same expected government input, we choose a relationship with equal integral size on the interval [0,1].

**Premium effects.**   The premium effect refers to the portion of a product's value that can outperform the market under normal operating conditions. As a company inputs more DI, the efficiency of the enterprise improves, resulting in a different premium value. By adjusting the product price, the enterprise can determine the premium value. However, similar to dynamic subsidies, existing research on premium effects assumes constant values [72]. To address this, we assume a non-constant dynamic form and apply the functional forms used in dynamic subsidies. Specifically, we set the positive correlation functional relations mentioned above as ($\alpha_1$, $\alpha_2$, $\alpha_3$) = $-v\gamma+k$, the quadratic relation form as ($\alpha_1$, $\alpha_2$, $\alpha_3$) = $-u\gamma^2+i\gamma$, and the primary terms as [($v_{\alpha1}$, $k_{\alpha1}$), ($v_{\alpha2}$, $k_{\alpha2}$), ($v_{\alpha3}$, $k_{\alpha3}$)]. The coefficients of the inverted U-shape are set as [($u_{\alpha1}$, $i_{\alpha1}$), ($u_{\alpha2}$, $i_{\alpha2}$),($u_{\alpha2}$, $i_{\alpha2}$)], and their values must satisfy the corresponding integration conditions.

## Updating rule optimization in an EDI diffusing network

**Fermi rule.**   With many digital enterprises in complex market networks, two strategies diffuse through the network connections. Initially, each enterprise adopts one pure strategy, and the proportion of enterprises in the network adopting the $D_i$ strategy (i.e., adopting the DI transformation strategy) is denoted by $x$, while those adopting the $D_a$ strategy (i.e., not adopting the DI transformation strategy) is $1-x$. At time $t$, an enterprise benefits from observing its own and its neighbors' strategies and gains at time $t-1$. Based on the Fermi rule (Fermi Dynamics), the individual playing the game obtains a game gain by interacting with all neighbors. When a firm $i$ decides to update its game strategy, it randomly selects one of its neighbors $j$ to compare payoffs, and the probability that firm $i$ will adopt neighbor $j$'s strategy in the next game is Eq (3) [42,73].

$$P_{(i \leftarrow j)} = \frac{1}{1 + e^{[u_i - u_j/k]}} \tag{3}$$

In this game, let $u_i$ and $u_j$ denote the gains obtained by individuals $i$ and $j$, respectively. Here, $k$ ($k \geq 0$) represents a noise effect that allows individuals to make irrational choices, implying that the strategy with lower gains still has a smaller probability of being adopted than other, high-yielding strategies. This strategy accounts for the possibility of errors in strategy adjustment under limited rationality. However, the random comparison of the return choices leads to greater return risk and uncertainty, and does not consider that each firm not only pursues the highest return strategy but also selects a less risky strategy. To optimize this strategy, we introduce the Savage regret value criterion in the following sections.

**Updating rule under the Savage criterion.**   Considering the fact that enterprises under the uncertain markets not only pursue maximum returns but also aim to reduce risks, this study introduces the Savage criterion to propose risk-averse evolutionary rules to optimize the existing network evolutionary criterion. In 1951, statistician Savage proposed the Savage criterion, also known as the minimax regret value criterion. This criterion is an essential tool in management and describes an approach taken by decision-makers to avoid greater opportunity losses in a state of nature without knowledge. The primary function of this criterion is to construct a regret value matrix, where each outcome entry represents a regret value defined as the difference between the best possible outcome and the given outcome. Subsequently, the matrix is used for risky decisions, whereby the expected regret values of other options are compared, and the smallest is selected as the factor determining the quality of the decision. The option corresponding to this value is the chosen option. The solution selected under this criterion allows for less risk while ensuring a larger benefit for the decision-maker.

The update rule employs the Savage criterion. First, the strategy chosen by each node in the next period at $t_0$ is selected based on the strategy of that node and the surrounding nodes in moment $t-1$ to select the strategy in moment $t$. Then, the Savage criterion is used for all the two possible strategies of the nodes at moment $t-1$ to decide the specific strategy at moment $t$. The steps involved are as follows:

**Step 1:** For any node in the network, represented by $w_i$, with an associated node $w_j$, the node has two strategies, $D_i$ and $D_a$. Based on the strategy chosen by the node at this time, the total gains $u_i$ and $u_j$ of the node and its linked node can be derived from the network game model.

**Step 2:** Assuming that the strategy chosen by the node at this moment is the opposite strategy, the total gains $u_i'$ and $u_j'$ of the node and its linked node can be determined.

**Step 3:** The difference between the total returns of this node and each of its associated nodes under the two strategies is found, and the maximum of the differences is taken. If the difference between the total return of the current strategy and that of the opposite strategy is greater, the strategy remains unchanged. However, if the opposite strategy yields a higher total return, the node's strategy is updated to the opposite strategy. At this point, the strategy at moment $t$ is determined as Eq (4)

$$D = \{(D_i, D_a) | \max(u_i - u_j, u_i' - u_j')\} \tag{4}$$

**Step 4:** The maximum value of the total gain difference of all linked nodes is compared to identify the node when the minimum value is obtained, and the strategy choice of this linked node is the strategy that $w_i$ node may imitate in the next moment. The Fermi update rule is then used to determine the $u_j$ value, which allows for the transformation of the updated probability is Eq (5)

$$P_{(i \leftarrow j)} = \frac{1}{1 + e^{[\min\max(u_i - u_j, u_i' - u_j')/k]}} \tag{5}$$

**Step 5:** Calculate the total return of the network game with the average return of the strategies.

## Simulation analysis and discussion

### Initial parameter setting

Drawing on existing research and empirical data [74,75], the complex network game model incorporates key parameters such as the cost of production for the DI technology. The model assumes a market consisting of 200 firms, with an initial proportion of 0.3 firms choosing to adopt the $D_i$ transformation strategy, and a noise effect of $k = 0.1$ in the update rule. Prior to $D_i$ strategy adoption, firms receive a product unit price of $f_a = 50,000$ yuan, a product demand quantity of $q = 5000$ units, other benefits of $P_i = 600,000$ yuan, and a labor cost of $C_i = 400,000$ yuan. To adopt the $D_i$ strategy, an input cost of $I = 500,000$ yuan is required, and premium factors of 0.2, 0.05, and 0.25 are assigned to the enterprise's product revenue, other revenue, and labor cost, respectively. The government subsidy is set at $b = 5000$ yuan, with a subsidy coefficient of $\eta = 1$. Within the market, the proportion of consumers with a preference for DI products is $\beta = 0.7$, while the proportion of consumers with a preference for traditional products is $1 - \beta = 0.3$. To account for complex inter-firm relationships, the model initializes network parameters such that each firm has five neighbors, and the probability of reconnection of broken edges is set to $p = 0.5$. The simulation of the model is based on 100 independent repetitions to ensure greater accuracy in the reported results. We employ a ring-form network of nodes to

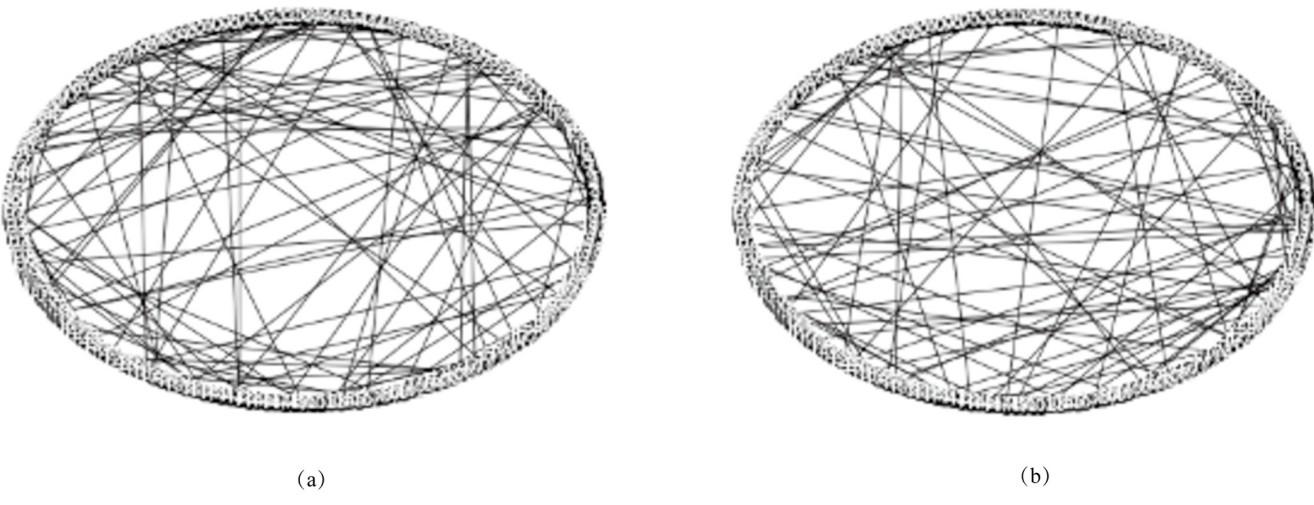

(a)                                                                                                          (b)

**Fig 2. Network linkage at $d$ = 3.**

clearly depict the connectivity of various nodes within the network. Note that we report the evolution of the strategies of all firms in the network with the overall benefits as a basis for measuring the final results [21].

## Comparison of diffusion results after optimization rules

To study the diffusion of EDI transformation when different neighbor nodes are optimized by updating rules, the number of linkages $d$ of each enterprise is set to $d$ = 3, $d$ = 4, and $d$ = 5, respectively. The network simulation results shown in Figs 2(A), 3(A) and 4(A) show the initial network linkages, and Figs 2(B), 3(B) and 4(B) show the network linkages at $T$ = 100 periods.

According to the above-established game model with two different update rules, the obtained diffusion results are shown in Figs 5–7.

The simulation results in Figs 5–7 demonstrate that for varying numbers of neighboring nodes ($d$ = 3, $d$ = 4, $d$ = 5), the diffusion results of DI transformation in the market are 0.2,

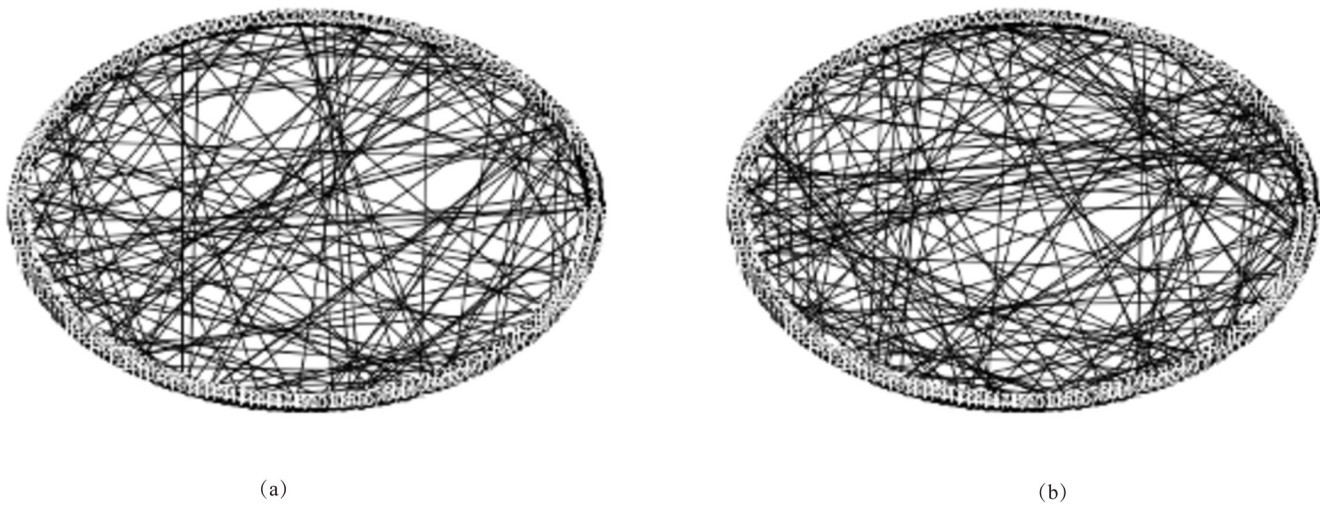

(a)                                                                                                          (b)

**Fig 3. Network linkage at $d$ = 4.**

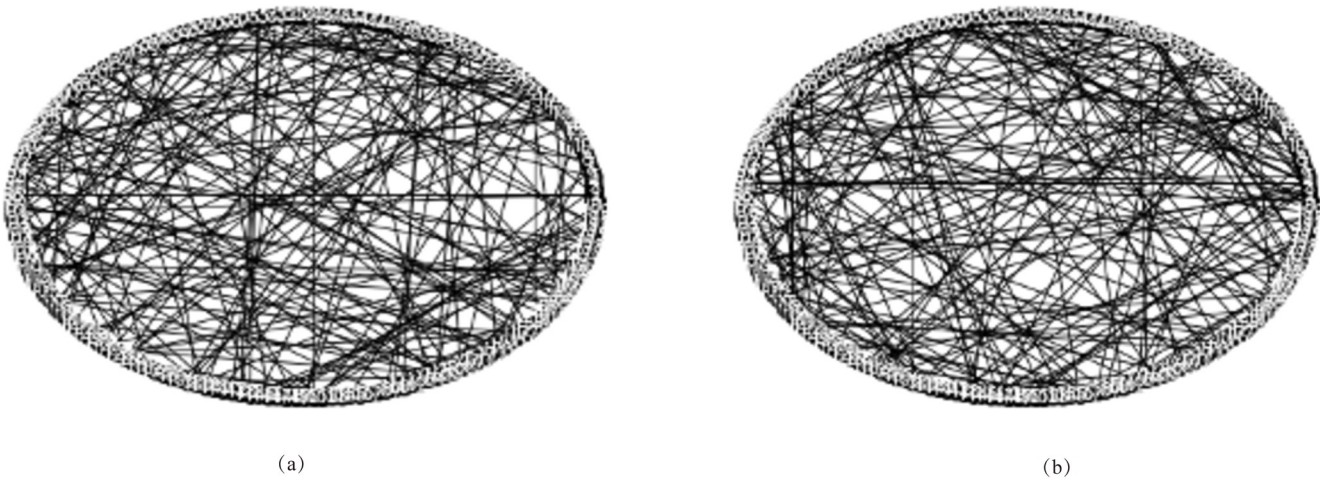

**Fig 4. Network linkage at $d$ = 5.**

0.42, and 0.5 with average gains of 7.74, 51.65, and 48.06 million yuan, respectively, when applying the unoptimized update rule. Similarly, when using the Savage criterion combined with the Fermi rule, the diffusion results of firms in the market with number-wise transformation are 0.73, 0.681, and 0.682, and the average gain is 35.54, 74.93, and 74.69 million yuan, respectively. Combining both sets of diffusion results with the average returns of enterprises shows that optimized strategy selection rules lead to greater average returns for firms adopting the $D_i$ strategy in the market, thereby promoting the number of enterprises undergoing this transformation. Market diffusion decreases and then increases with the number of neighbors, while the average revenue of the firm increases and then decreases. These findings suggest that only the right number of neighboring firms can achieve optimal outcomes for both the market and the firm. To explore more complex market enterprise network situations, the number of enterprise neighbors $d$ is fixed at 5 for subsequent analyses.

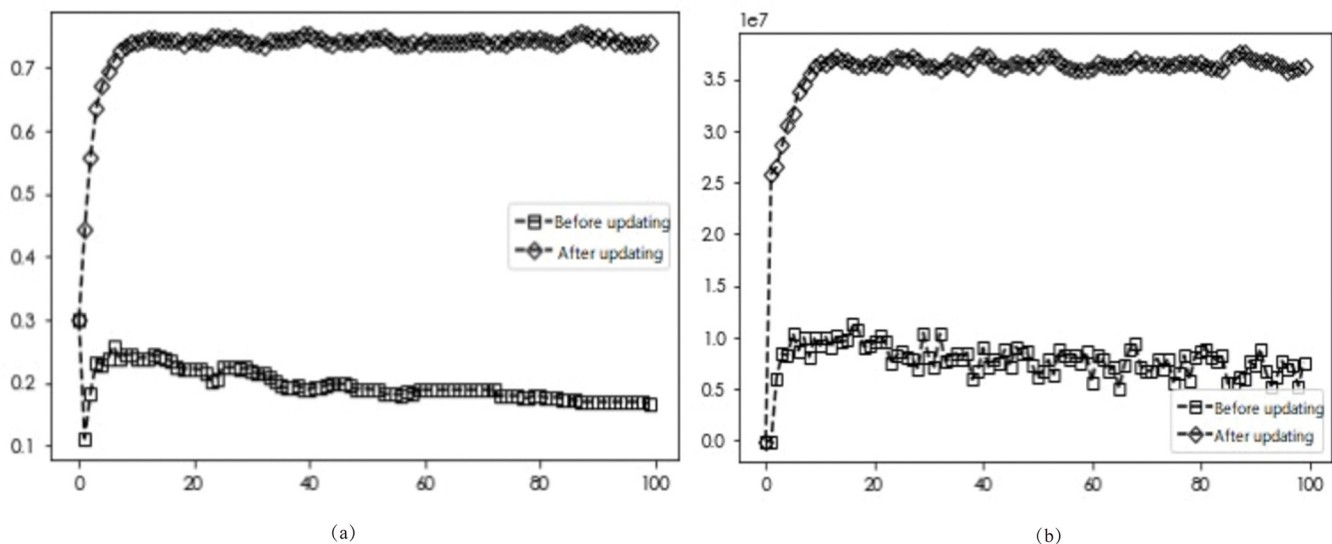

**Fig 5. Simulation results before and after the optimized update rule for $d$ = 3.**

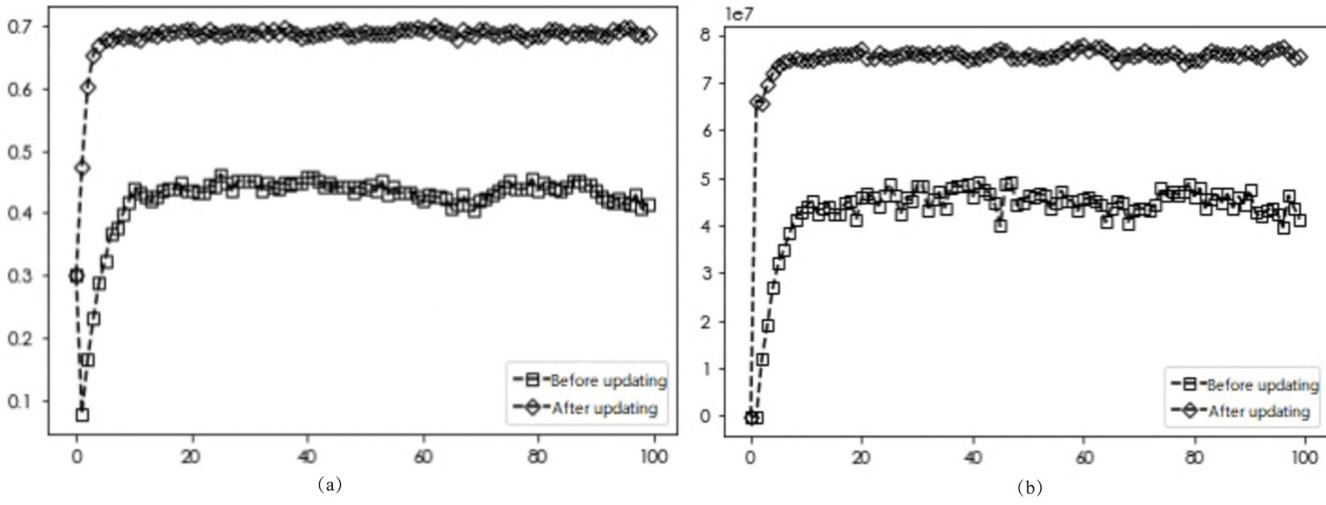

**Fig 6. Simulation results before and after the optimized update rule for $d = 4$.**

## Dynamic subsidies

Fig 8 illustrates the diffusion and average return of DI transformation for enterprises when dynamic subsidies are introduced. The dynamic subsidy has a significant impact on the diffusion of DI transformation, with the best diffusion obtained when using dynamic non-linear subsidies (0.685, 0.682, 0.690). As indicated in **Fig 8(B)**, under dynamic subsidies, linear subsidies result in lower benefits for firms with DI transformation than those under static subsidies. Conversely, inverted U-shaped subsidies lead to higher benefits for firms with DI transformation than those under static subsidies. These findings suggest that under different subsidy methods, although the number of government subsidies remains the same, different subsidy forms can yield varying benefits for DI transformation of enterprises, with the inverted-U type subsidy approach yielding the best results.

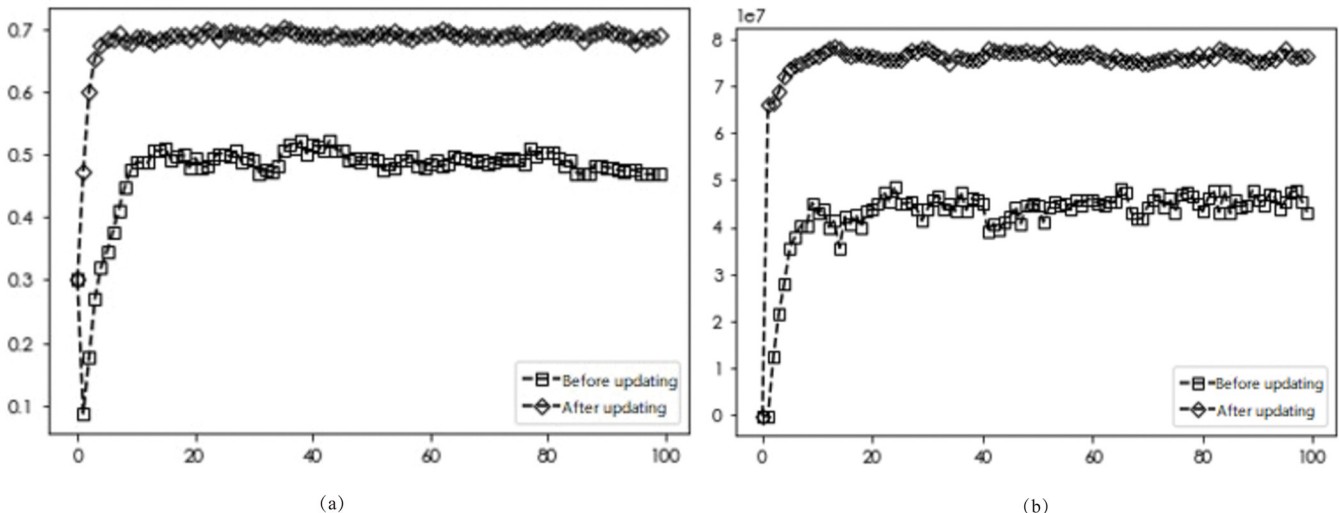

**Fig 7. Simulation results before and after the optimized update rule for $d = 5$.**

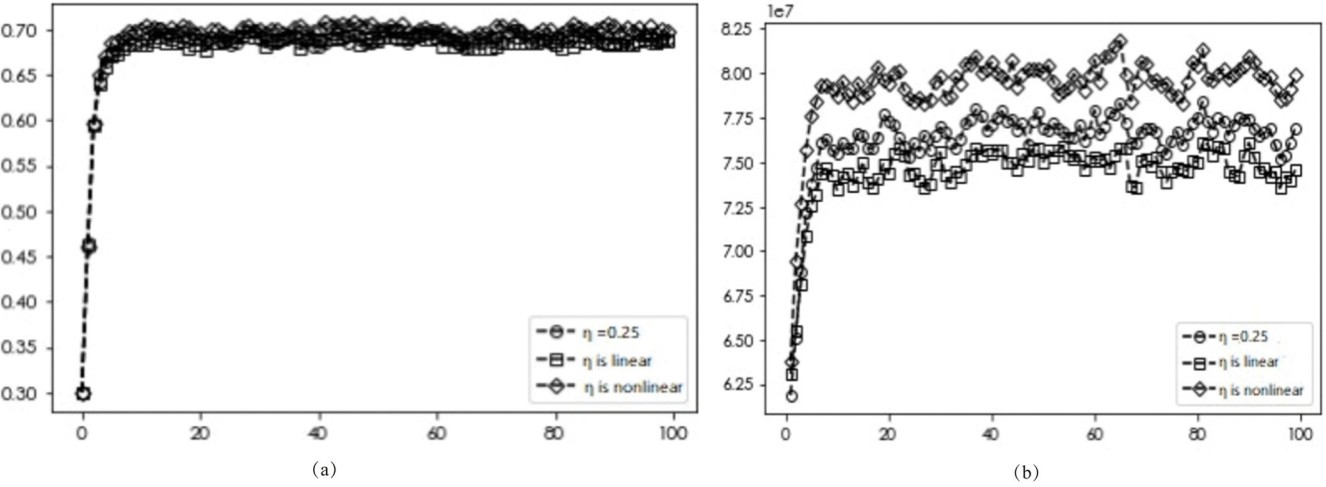

**Fig 8. Simulation results under different subsidy schemes.**

Specifically, the static, linear dynamic, and nonlinear dynamic subsidies proposed in this study are all strategies that the government can adopt to subsidize policies, with the more complex nonlinear dynamic subsidies usually obtaining better results [76]. However, the difficulty in implementing government subsidy-related policies is associated with the complexity of technology diffusion networks. Considering the difficulty of implementing technology subsidies and the degree of realism, research based on the cost of policy implementation in this case and the degree of cooperation of the public found that the implementation of a general static subsidy strategy can also achieve better results [77,78].

## Premium effects

The diffusion results and average returns of EDI transformation under varying premium effects are presented in Figs 9–11. Figs 9 and 10 reveal a consistent diffusion result of 0.682, irrespective of the premium effects on other benefits and labor costs, indicating that different

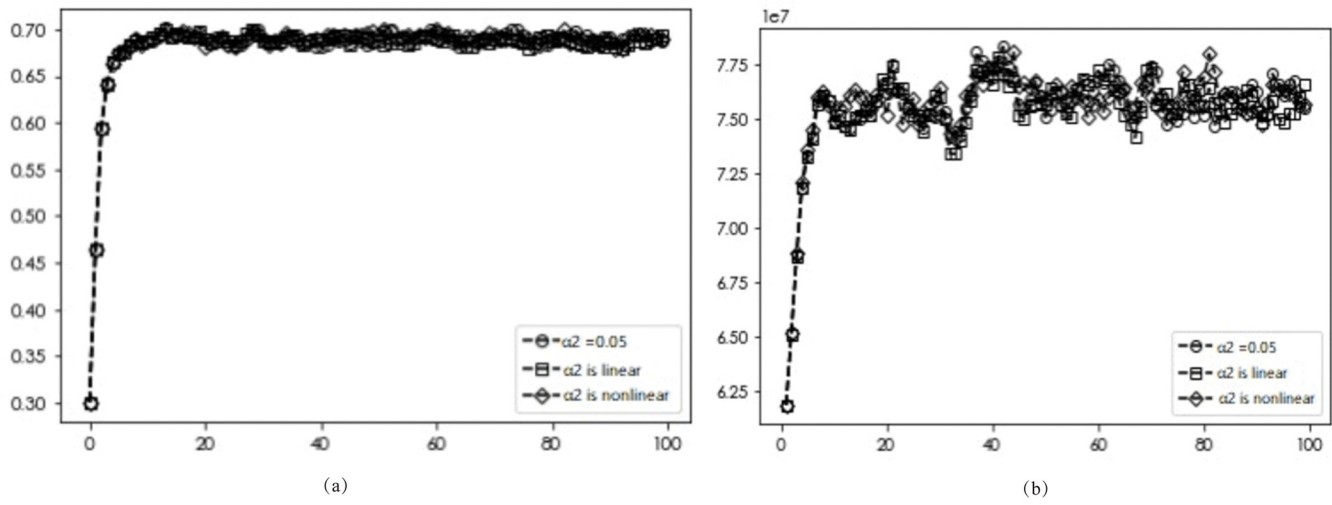

**Fig 9. Simulation results with different other revenue premiums.**

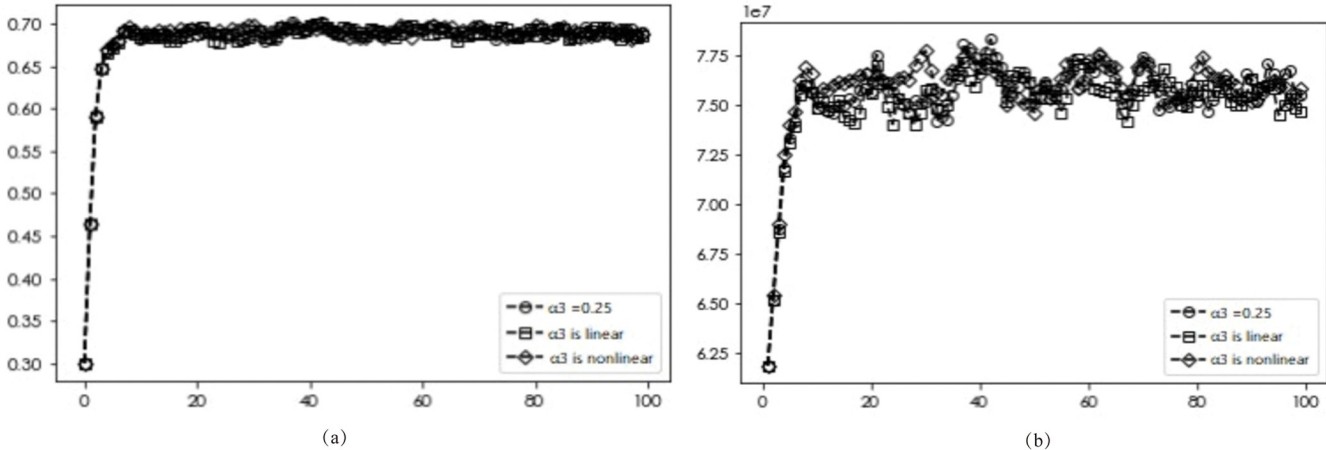

**Fig 10. Simulation results under different labor cost premiums.**

forms of premium effects on these factors will not affect the final diffusion results. In the case of the product premium effect, Fig 11 shows that the optimal diffusion scenario for DI transformation can be achieved when the dynamic non-linear pricing approach is used (0.682, 0.673, 0.697).

Further analysis, as depicted in Fig 11(B), reveals that for dynamic premium, linear pricing forms result in lower returns for enterprises with DI transformation than those under static subsidies, while inverted U-shaped pricing forms lead to higher returns for enterprises with DI transformation than those under static pricing. These findings suggest that under different pricing methods, although the pricing ratio remains the same, different pricing forms can yield varying benefits for EDI transformation, with the inverted U-shaped pricing form yielding the best results. Specifically, enterprises should opt to gradually increase product prices in the early stages of DI product sales, and subsequently reduce prices after the digital and intellectual enterprise has gained a certain market share to obtain higher benefits. Such price adjustments can further increase the share of digital and intellectual enterprises within the market.

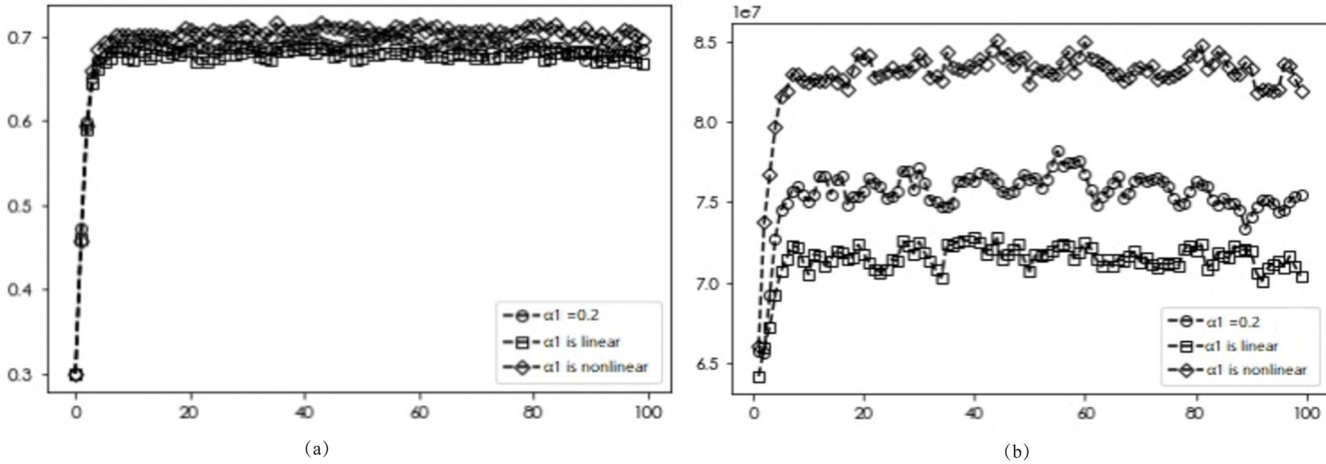

**Fig 11. Simulation results under different DI product premiums.**

The pricing of an enterprises' products should be within a certain threshold, and different pricing sequences can bring different benefits. Enterprises can continuously adjust their pricing regarding market consumer preferences and policy support. According to the principle of differentiated pricing, the pricing form of "low-high-low" in the entire life cycle of the product (i.e., shelf, growth, maturity, and decline stages) is consistent with the demand curve of the commodity and the law of value and with consumers' expectations of new product pricing [79].

## Conclusions and implications

### Conclusions

This research provides strategies and rationale for EDI transformation decisions by considering market demand and government policy coordination, and also complements existing research methods on EDI transformation. We systematically reveal the intrinsic evolution mechanism of EDI transformation through a complex network evolution game model. On the basis of different strategy choices, the article provides a more appropriate strategy for firms that need to transform under uncertain markets by improving the updating rules. In addition, the article seeks a more appropriate strategy for firms and the government itself by comparing the diffusion results under fixed, linear, and nonlinear scenarios of subsidies and premiums. Our study findings are as follows:

1. For an enterprise, the strategy that balances risk minimization and benefit maximization leads to better returns and is more capable of stimulating the diffusion effect of digitization in the market. Cooperation with the appropriate number of neighboring enterprises can better promote the transformation of EDI.

2. The government can promote the transformation of DI through incentives. Specifically, a reasonable and dynamic subsidy policy has a better effect on promoting the transformation of DI, whereas an unreasonable dynamic subsidy hinders the process. In this study, we show that the inverted U-shaped dynamic balance policy is better than other policies, and the subsidy method ensures that the expected promotion effect can be achieved under the same expected input.

3. Under the influence of consumer preferences, enterprises can adjust the pricing of DI products to influence their degree of DI transformation. Static and dynamic pricing schemes can be adopted, and the inverted U-type pricing scheme can achieve maximum revenue while promoting other neighboring enterprises in the market of DI transformation.

### Implications

Enhancing the efficiency of EDI transformation is crucial to China's economic development in the digital era. This study not only provides a different perspective from the existing literature for analyzing the diffusion of EDI transformation but also provides ideas on ways to formulate decision-making behaviors related to government subsidies and enterprise pricing under environmental regulation. To promote the transformation of EDI and realize China's vision of socialist modernization, the following suggestions are proposed based on this study and existing empirical evidence. (1) Adjust the government subsidy program. The government should implement dynamic subsidy policies according to the market situation and establish special support programs for EDI transformation. This will enhance the initiative and enthusiasm for EDI transformation. Financial institutions should also increase their support for EDI transformation and guide enterprises to invest in DI technology. The government can use various types of publicity media to convey the premium effect of DI to enterprises, changing the co-

competition relationship between enterprises in the market and promoting the transformation of EDI from the "demand side." (2) Improve enterprise technology premiums and reasonable pricing of products. Enterprises should combine DI with basic technology innovation, promote artificial intelligence and key technology breakthroughs, enhance data analysis capabilities, and improve manufacturing facilities and manpower to achieve accurate and dynamic adjustment of operational efficiency. By combining market share, product advantages, and consumer preferences, enterprises can make reasonable pricing for DI products, improve market competitiveness, and promote the transformation of DI. (3) Enhance consumers' awareness of DI. Enterprises must create more experience opportunities for consumers to interact with DI products, develop new customers and discover potential customers, and absorb a large number of potential users to form continuous and stable consumers of DI products. Enterprises should also find the appropriate number of neighboring enterprises for DI cooperation, systematically create an internal communication platform to enhance data exchange-ability among enterprises, give full play to the industrial cluster effect, and accelerate the DI transformation process of enterprises adopting DI strategy in the market while improving their income.

This study has certain limitations. The selection of the sub parameters in the TOE framework has some limitations and can be further combined with questionnaires or case studies, which can enhance the richness of the framework. Meanwhile, since the research only exists in the game at the enterprise level, multi-layer complex networks can be added for analysis in the future. Furthermore, this study does not carry out empirical research based on the factors influencing transformation, and this can be addressed in future research using questionnaire data.

## Supporting information

**S1 Data.**
(ZIP)

## Author Contributions

**Conceptualization:** Huan Hu.

**Formal analysis:** Mingyu Zhao.

**Methodology:** Huan Hu.

**Project administration:** Huan Hu.

**Software:** Xiaoyi Zhang.

**Validation:** Huan Hu.

**Visualization:** Xiaoyi Zhang.

**Writing – original draft:** Mingyu Zhao.

**Writing – review & editing:** Mingyu Zhao.

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
