## [Decision Letter · Decision Letter 0]

26 May 2023

PONE-D-23-06664Diffusion and Evolution of Enterprise Digitalization and Intelligentization (EDI) Transformation and Upgrading: A TOE Framework-based AnalysisPLOS ONE

Dear Dr. HU,

Thank you for submitting your manuscript to PLOS ONE. After careful consideration, we feel that it has merit but does not fully meet PLOS ONE’s publication criteria as it currently stands. Therefore, we invite you to submit a revised version of the manuscript that addresses the points raised during the review process.

We look forward to receiving your revised manuscript.

Kind regards,

Donato Morea, Ph.D.

Academic Editor

PLOS ONE

Journal Requirements:

Reviewers' comments:

Reviewer's Responses to Questions

**Comments to the Author**

1. Is the manuscript technically sound, and do the data support the conclusions?

Reviewer #1: Partly

Reviewer #2: Yes

2. Has the statistical analysis been performed appropriately and rigorously? 

Reviewer #1: No

Reviewer #2: N/A

3. Have the authors made all data underlying the findings in their manuscript fully available?

Reviewer #1: No

Reviewer #2: No

4. Is the manuscript presented in an intelligible fashion and written in standard English?

Reviewer #1: Yes

Reviewer #2: Yes

5. Review Comments to the Author

Reviewer #1: The topic of this manuscript is very interesting, but many key factors are not taken into account in the paper. I do not agree with the research framework of Figure1 in the manuscript. Why is the technical aspect employee cost and business level? Why not consider digital technology versus existing technology? Why on the organizational side is R&D investment and technology premium?

Reviewer #2: This paper mainly investigates how to promote the process of EDI by complex network game theory and TOE framework. This is a hot topic problem nowadays. However, it’s a pity that the literature contribution of this version manuscript is relatively marginal. Some detailed advice is listed as follows.

(1) You analyze the three levels of TOE technology, organization, and environment, and derive the payoff matrix providing a theoretical basis for the game model. However, the relationship between them seems very ambiguous.

(2) In line 175, what’s the meaning of other income (such as production and operation efficiency improvement)?

(3) EDI is an accumulation process in reality. The complex network game in this paper now can’t reflect the feature. Meanwhile, Hypothesis 2 seems not correspond to the fact of reality. As such, some analysis based on real situations should be given.

(4) Some notations haven’t been explained clearly. Such as the equation in line 263. Hence, you’d better give a notation table to make them. Meanwhile, some terms are not explained clearly, e.g., ‘premium effect’.

(5) Typos in Table 1. It is ‘Not Adopt’ instead of ‘Adopt’.

Finally, except for the main problems mentioned above, there are still some typos and mistakes in this paper. You need to check out carefully.

6. PLOS authors have the option to publish the peer review history of their article (what does this mean?). If published, this will include your full peer review and any attached files.

Reviewer #1: No

Reviewer #2: No

---

## [Author Response · Author response to Decision Letter 0]

28 Jun 2023

Reviewer #1

Q1: The topic of this manuscript is very interesting, but many key factors are not taken into account in the paper. I do not agree with the research framework of Figure1 in the manuscript. Why is the technical aspect employee cost and business level? Why not consider digital technology versus existing technology? Why on the organizational side is R&D investment and technology premium?

Answer: We feel great thanks for your professional review work on our article.

First, we were really sorry for our careless mistakes. In Figure 1, the technical and organizational factors are placed upside down. In that research framework of fig. 1, the correct writing should be that the technical aspect includes R&D investment and technology premium, and the organizational aspect includes staff cost and business level. The correct picture has been updated in line 193 of the manuscript.

According to your nice suggestions, we sorted out some research related to the influencing factors of digital transformation. We classify these influencing factors according to the TOE framework. The classification results are shown in Table 1. (The table has been added to the text of the manuscript)

As can be seen from Table 1, the technical aspects mainly include digital technology, AI technology and R&D. In our manuscript, with reference to Ancillo and Gavrila (2023) and Zapata et al. (2021), the first factor in technology selection is R&D investment. The second factor is the technology premium. Digital technology and artificial intelligence technology, these two words are widely used. Traditional technology mainly uses computers, networks and other hard equipment software to innovate, and its application scope has certain limitations. Digital technology is the general name of artificial intelligence, big data, block chain and other scientific technologies. Compared with traditional technologies, digital technology is more intelligent. The leading enterprise's digital intelligence transformation provides a demonstration for the enterprises in the digital intelligence transformation stage, and the enterprises in the digital intelligence transformation stage will learn from and learn from new digital technologies and business models to improve their production and operation efficiency. This process is summarized as technology premium. Technology premium, the word describes the technical advantages of digital technology itself and the influence of digital technology on traditional technology.

The Organization aspects mainly include financial resources, business level and human capital. In our manuscript, we choose staff cost as one of the organizational factors. Because staff cost can not only reflect human capital, but also reflect the characteristics of financial resources. Another factor, referring to the research of Benlian and Haffke (2016), Hansen et al (2011), Singh and Hess (2017), Jackson and Dunn-Jensen (2021), chooses business level.

Reviewer #2

This paper mainly investigates how to promote the process of EDI by complex network game theory and TOE framework. This is a hot topic problem nowadays. However, it’s a pity that the literature contribution of this version manuscript is relatively marginal. Some detailed advice is listed as follows. 

Q1: You analyze the three levels of TOE technology, organization, and environment, and derive the payoff matrix providing a theoretical basis for the game model. However, the relationship between them seems very ambiguous. 

Q2: In line 175, what’s the meaning of other income (such as production and operation efficiency improvement)? 

Q3: EDI is an accumulation process in reality. The complex network game in this paper now can’t reflect the feature. Meanwhile, Hypothesis 2 seems not correspond to the fact of reality. As such, some analysis based on real situations should be given. 

Q4: Some notations haven’t been explained clearly. Such as the equation in line 263. Hence, you’d better give a notation table to make them. Meanwhile, some terms are not explained clearly, e.g., ‘premium effect’. 

Q5: Typos in Table 1. It is ‘Not Adopt’ instead of ‘Adopt’.

We feel great thanks for your professional review work on our article.

According to your nice suggestions, we further clarify the contribution of this article. Firstly, it focuses on the micro subjects of Digitalization and Intellectualization Enterprises (DIE), deriving the conditional logic of EDI transformation based on the TOE framework of technological innovation diffusion. This research is a useful attempt to apply the toe framework to the transformation of enterprise digital intelligence, which broadens the application boundary of the toe framework. Secondly, the study addresses the limitations of the Fermi update rule in different game situations of the complex network of EDI. By combining the savage criterion to design a renewal strategy, the study aims to improve the enterprise returns based on risk reduction. This research provides decision-making basis for enterprises to improve product pricing. Finally, the study introduces an inverted U-shaped subsidy function and a dynamic subsidy to the traditional evolutionary game model to compare the advantages of different government subsidy strategies. This approach provides a new reference for the government to promote DI reform in China. We added this part to lines 64~74.

Answer 1: Thank you very much for your question. We realize that in the manuscript, we focus on the single aspect of technology, organization and environment, but the relationship among them is vague.

Technology is the foundation of EDI. Enterprises that implement EDI form technological advantages by increasing R&D investment, and provide consumers with alternative and innovative product choices. The spillover of digital technology has had a demonstration effect on enterprises that implement traditional technology.

Organization is the main body of EDI. Staff resources and financial resources are both important factors in an organization, and staff cost takes both into account. The integration of digital technology into existing organizations requires employees who are proficient in digital technology, and the participation of digital labor force will bring about the improvement of management level.

The relationship between the three can be summarized as such a passage. Policies in the environment will guide the technological and organizational progress of enterprises. The competitive relationship in the environment will force enterprises that implement traditional technologies to improve their technical level and improve their organizational structure. The cooperative relationship in the environment can stimulate the positive effect of technology spillover and accelerate the progress of the organization. Technological progress is guided by policies in environmental factors and stimulates the industry environment to form a learning effect. The progress of the organization has created a peer effect for the industry environment.

We added this part to lines 182~189. And we integrate the relationship between the three into the research framework of Fig 1, which has been further improved.

Answer 2: In line 175, the specific meaning of other income in the manuscript refers to all economic income except the income from main business. These benefits are formed through daily activities such as selling goods, providing labor income and transferring the right to use assets. These benefits have the characteristics of low frequency, small amount of each business and low proportion of income. In addition, we accurately define the product income mentioned in the manuscript as the main business income (mainly through products) that enterprises can obtain by adopting traditional digital means. It has been supplemented in the description of important parameters in Table 2.

Answer 3: EDI is an accumulation process in reality. The income of complex network proposed in this paper is the income of all enterprises in the whole network. In the process of network game, it gradually realizes stable growth, which reflects that EDI achieves stable income in the process of continuous accumulation. Take the dynamic subsidy as an example. As shown in Figure 8a, under the dynamic subsidy, the proportion of EDI is 30%, and after the dynamic subsidy adjustment, the proportion of EDI has accumulated to 70%. At the same time, as shown in figure 8b, corporate income is also growing.

In Hypothesis 2, we use the specific case of new energy vehicles to further elaborate the hypothetical content. According to the theory of consumer preference, consumer behavior is determined by their preferences. When facing the choice between traditional and digital intelligent products in the market, the profits generated by different enterprise strategies can vary depending on consumers' preferences. For instance, in the early days of the launch of new energy vehicles, due to immature technology, most consumers still preferred traditional products, and traditional vehicles remained the main source of sales profits for enterprises. However, with the continuous development of digital and intelligent technologies, such as intelligent driving, planning, key element perception, and energy saving, automobile consumers' preferences have shifted towards digital intelligence. These consumers are willing to pay higher prices for products with more digital intelligence characteristics. This shift has resulted in a significant increase in the proportion of sales profits from new energy vehicles in enterprises, which has further incentivized the automobile industry to prefer the production of digital intelligent products. We added this part to lines 240~255.

Answer 4: Important parameter Table 2 has been added to the manuscript. Compared with traditional technologies, digital technology is more intelligent. The leading enterprise's digital intelligence transformation provides a demonstration for the enterprises in the digital intelligence transformation stage, and the enterprises in the digital intelligence transformation stage will learn from and learn from new digital technologies and business models to improve their production and operation efficiency. This process is summarized as technology premium. For the premium effect, α1 is the premium factor of the product income of the digital intelligent transformation and upgrading enterprise (referring to the fact that the production of digital intelligent products can improve the corresponding product income in the process of digital intelligent transformation). α2 is the premium factor for other benefits of enterprises in digital intelligent transformation and upgrading (it means that other benefits can be improved by adopting digital intelligent technology and management means in the process of digital intelligent transformation). α3 is the premium factor for the labor cost of enterprises in the process of digital intelligence transformation and upgrading (it means that adopting digital intelligence technology and management means can reduce the corresponding labor cost in the process of digital intelligence transformation).

Answer 5: Thanks for your careful checks. We are sorry for our carelessness. Based on your comments, we modified the word ‘Not Adopt’ in Table 1 and checked the whole manuscript.

---

## [Decision Letter · Decision Letter 1]

11 Oct 2023

PONE-D-23-06664R1Diffusion and Evolution of Enterprise Digitalization and Intelligentization (EDI) Transformation and Upgrading: A TOE Framework-based AnalysisPLOS ONE

Dear Dr. HU,

Thank you for submitting your manuscript to PLOS ONE. After careful consideration, we feel that it has merit but does not fully meet PLOS ONE’s publication criteria as it currently stands. Therefore, we invite you to submit a revised version of the manuscript that addresses the points raised during the review process.

We look forward to receiving your revised manuscript.

Kind regards,

Donato Morea

Academic Editor

PLOS ONE

Reviewers' comments:

Reviewer's Responses to Questions

**Comments to the Author**

1. If the authors have adequately addressed your comments raised in a previous round of review and you feel that this manuscript is now acceptable for publication, you may indicate that here to bypass the “Comments to the Author” section, enter your conflict of interest statement in the “Confidential to Editor” section, and submit your "Accept" recommendation.

Reviewer #1: (No Response)

Reviewer #2: All comments have been addressed

2. Is the manuscript technically sound, and do the data support the conclusions?

Reviewer #1: (No Response)

Reviewer #2: Partly

3. Has the statistical analysis been performed appropriately and rigorously? 

Reviewer #1: (No Response)

Reviewer #2: N/A

4. Have the authors made all data underlying the findings in their manuscript fully available?

Reviewer #1: (No Response)

Reviewer #2: (No Response)

5. Is the manuscript presented in an intelligible fashion and written in standard English?

Reviewer #1: (No Response)

Reviewer #2: (No Response)

6. Review Comments to the Author

Reviewer #1: The reviewer believes that the topic “Diffusion and Evolution of Enterprise Digitalization and Intelligentization (EDI) Transformation and Upgrading: A TOE Framework-based Analysis” is worthy of investigation. However, the following needs to be addressed. There are minor and major issues that should be corrected. I believe the paper could be further strengthened by added information about.

What is the purpose of the paper? What are the research implications?

Please reorganize the manuscript at the journal request. Please change the reference format.

The language of this manuscript is very bad and needs help from native speakers.

The title of the manuscript should fully demonstrate the content of this study and the relevant subjects.

Abstracts should include the purpose and findings of the study. Please simplify the presentation of the study process.

According to data released....................... This a very vague statement. These sentences do not provide any information on how the concept could be conceptualized?

Building on the preceding analysis.................... This section should explain the study's context and research objective. Furthermore, the research gap needs to be narrowed after analyzing the previous studies. The research method is not adequately explained in the first section.

Here author must build research gap following the previous studies.-The manuscript does not answer the following concerns: Why is it timeliness to explore such a study? What makes this study different from the previously published studies? Are there any similarly findings in line with the previously published studies? Are the findings different from prior academic studies that were conducted elsewhere, if any? What are the new technologies, some recent issue highlights the importance. See: An adoption-implementation framework of digital green knowledge to improve the performance of digital green innovation practices for industry 5.0. https://doi.org/10.1016/j.jclepro.2022.132608

-There is no flow in the text. It partly depends on the lack of proofreading but also on the fact that many statements and claims are made without being followed up by a clear and logical discussion. It is especially problematic in the Introduction that brings up a number of findings from different areas without linking them together.

-More importantly, the choice of the questionnaire questions should be explained in light of the theory and the prior literature on the topic. The arguments are simply relationships and causes very close to the replication of many studies dealing with the same thing. For example, what is connection of upgrading path of manufacturing enterprises from the value perspective. See the following: The governance mechanism of the building material industry (BMI) in transformation to green BMI: The perspective of green building. https://doi.org/10.1016/j.scitotenv.2019.04.317

-Methodology: Model.. I suggest authors here build your main heading on Research and data methodology. Clearly explain the model building process, and what previous studies have used similar models (model testing approach).

See the following: A three-player game model for promoting the diffusion of green technology in manufacturing enterprises from the perspective of supply and demand.https://doi.org/10.3390/math8091585

A stochastic differential game of low carbon technology sharing in collaborative innovation system of superior enterprises and inferior enterprises under uncertain environment, https://doi.org/10.1515/math-2018-0056

Information fusion for future COVID-19 prevention: continuous mechanism of big data intelligent innovation for the emergency management of a public epidemic outbreak.

Please consider this structure for manuscript final part.-Discussion-Conclusion-Managerial Implication-Practical/Social Implications

Please make sure your conclusions' section underscores the scientific value-added of your paper, and/or the applicability of your findings/results. Highlight the novelty of your study. In addition to summarizing the actions taken and results, please strengthen the explanation of their significance. It is recommended to use quantitative reasoning comparing with appropriate benchmarks, especially those stemming from previous work.

Reviewer #2: (No Response)

7. PLOS authors have the option to publish the peer review history of their article (what does this mean?). If published, this will include your full peer review and any attached files.

Reviewer #1: No

Reviewer #2: No

---

## [Author Response · Author response to Decision Letter 1]

16 Nov 2023

We sincerely thank the editor and all reviewers for their valuable feedback that we have used to improve the quality of our manuscript. At the same time, we revised the article according to the format requirements and template of your journal, and added corresponding attachments and public data links for data, codes and pictures as required. The reviewer comments are laid out below in italicized font and specific concerns have been numbered. In addition, relevant literature for which the reviewers provided references was cited in the suitable place. Our response is given in normal font and changes/additions to the manuscript are given in revision mode. At the end of the reply letter, the article needs to add financial support, the specific content is given in the reply, and the relevant content is also added in the cover letter.

Reviewer #1: 

The reviewer believes that the topic “Diffusion and Evolution of Enterprise Digitalization and Intelligentization (EDI) Transformation and Upgrading: A TOE Framework-based Analysis” is worthy of investigation. However, the following needs to be addressed. There are minor and major issues that should be corrected. I believe the paper could be further strengthened by added information about.

Q1: What is the purpose of the paper? What are the research implications?

Answer: We thank the reviewers for your comments, according to which the enterprise digitalization and intelligence (EDI) studied in the article is an important aspect of China's modernization process. “Given that uncertainty in government and business decisions hinders EDI transformation, research on the evolutionary process of EDI diffusion and the strategies adopted by stakeholders in market environments is necessary. Therefore, … The findings of the study can provide valuable insights into factors that drive EDI transformation”. The paragraph has been added to the introduction.

Q2: Please reorganize the manuscript at the journal request. Please change the reference format.

Answer: Thanks to the reviewer's comments, the article has been reformatted to follow the journal format and the appendix images, data, and code files have been updated.

Q3: The language of this manuscript is very bad and needs help from native speakers.

Answer: We thank the reviewers for your comments, based on which we sought the help of native speakers and revised the language of the article, as shown in the marked section of the revised paper with proof of touch-ups.

Q4: The title of the manuscript should fully demonstrate the content of this study and the relevant subjects.

Answer: We thank the reviewers for your comments, and based on the review comments, the title has been revised to take into account the content and topic of the study: Network evolution of diffusion in enterprise digitalization and intellectualization transformation: A technology–organization–environment framework perspective.

Q5: Abstracts should include the purpose and findings of the study. Please simplify the presentation of the study process.

Answer: Thanks to the reviewer's comments, according to the reviewer's comments, we revised and added the purpose and significance of the study to the abstract of the article. At the same time, we have simplified the description of the research process by revising the abstract to read “Enterprise digitalization and intellectualization (EDI) is a crucial aspect of China’s modernization process. However, uncertainty in market and business decisions hinders the EDI diffusion process in China … They also indicate recommendations for government policies to enhance diffusion efficiency and reasonable pricing for enterprises to promote returns”. For specific changes, please see the markup in the abstract section of the revised manuscript.

Q6: According to data released....................... This a very vague statement. These sentences do not provide any information on how the concept could be conceptualized?

Answer：Thanks to the reviewer's comments, according to the reviewer's comments, the data at this place in order to lead to the importance of the subsequent concept of digital intelligence, so the following modification has been added after the paragraph as a supplement “Data captured by digital technology needs to be further explored for its value through the application of digital intelligence technology, thus, there is an urgent need to develop concepts other than digitization.” For specific changes, please see the paragraph markers in the revised manuscript.

Q7: Building on the preceding analysis.................... This section should explain the study's context and research objective. Furthermore, the research gap needs to be narrowed after analyzing the previous studies. The research method is not adequately explained in the first section.

Answer：Thanks to the reviewer's comments, according to the reviewer's comments, we added the related research on digital intelligence in the introduction part, and further added the research on the gaming methods related to the EDI transformation and its shortcomings in that part. Meanwhile, in the first part, we add a description of the relevant research methods and propose some methodological gaps in the research. On this basis, we briefly explain the background and purpose of the study in this paragraph. Specific changes are marked in the revised manuscript.

Q8: Here author must build research gap following the previous studies. -The manuscript does not answer the following concerns: Why is it timeliness to explore such a study? What makes this study different from the previously published studies? Are there any similarly findings in line with the previously published studies? Are the findings different from prior academic studies that were conducted elsewhere, if any? What are the new technologies, some recent issue highlights the importance. See: An adoption-implementation framework of digital green knowledge to improve the performance of digital green innovation practices for industry 5.0. https://doi.org/10.1016/j.jclepro.2022.132608

Answer: Thanks to the reviewer's comments, according to the reviewer's comments and with reference to related studies, the relevant content in the article has been enriched based on the following considerations: the article is considered to address is to further propose the concept of digitization, and at the same time to study the impacts of different corporate decisions in the market during the process of digital and intelligent transformation of enterprises. The study has similarities with existing studies, i.e., both consider changes in enterprise transformation under digital or intelligent technologies, but the difference is that the existing studies only unilaterally consider digital or smart transformation, with fewer references to the new concept and technology of DI. Meanwhile, for the application of the method, most of the articles adopt the method of two-party or multi-party evolutionary game, without considering the results of the game after the formation of complex networks by different enterprises. The main parts that need to be revised have been mentioned in the introduction, and of course, the relevant parts of other parts of the article have been revised, such as the abstract and the conclusion, and the specific changes are marked in the revised manuscript.

Q9: There is no flow in the text. It partly depends on the lack of proofreading but also on the fact that many statements and claims are made without being followed up by a clear and logical discussion. It is especially problematic in the Introduction that brings up a number of findings from different areas without linking them together.

Answer: Thanks to the reviewer's comments, according to the reviewer's comments, the article organized the logic of the introduction part, according to the logic of "research background - digitalization or intelligence research - DI research - EDI transformation research - research deficiencies - existing research methods - research deficiencies - research objectives - research significance" modified the introduction and added part of the literature enriched the relevant content. At the same time, each part is organized according to the depth of the literature research content, and the specific changes are marked in the introduction of the revised manuscript.

Q10: More importantly, the choice of the questionnaire questions should be explained in light of the theory and the prior literature on the topic. The arguments are simply relationships and causes very close to the replication of many studies dealing with the same thing. For example, what is connection of upgrading path of manufacturing enterprises from the value perspective. See the following: The governance mechanism of the building material industry (BMI) in transformation to green BMI: The perspective of green building. https://doi.org/10.1016/j.scitotenv.2019.04.317

Answer: Thanks to the reviewer's comments, according to the reviewer's comments and with reference to related studies, we have made the following considerations: “First, at the technical level, technology R&D investment will enhance the degree of specialization of enterprises for the application of DI technology. Simultaneously, the research and development of new technologies will also enable the enterprise to enhance the corresponding knowledge base, and accumulate knowledge of DI transformation. The technology premium effect has a positive impact on the economic consequences of product returns, production, and operational efficiency, and the economic foundation determines the superstructure, which can increase investment in the research and development of digital intelligence technology and the introduction of relevant talent with the support of a sufficient economy. Second, at the organizational level, human resources play a crucial role in promoting EDI transformation. Human resources are an important cost for enterprises, which can not only promote the transformation of knowledge results generated by R&D investment, but may also reduce labor costs via the introduction of DI technology. Third, at the environmental level, enterprises are not only in a competitive relationship with similar firms but also affected by heterogeneous consumers in the marketplace and government policies. Consumers’ consumption concepts advance the important driving force of EDI transformation, which can source operating funds as well as enhance the enterprise’s reputation and promote the sustainable development of the enterprise. Simultaneously, only by adhering to government policies can we better seize opportunities in the international environment, realize the social value of enterprises, and meet the needs of national economic development”. The paragraph has been added to the second subsection of Part 2 with relevant references, as indicated in the revised manuscript marker.

Q11: Methodology: Model. I suggest authors here build your main heading on Research and data methodology. Clearly explain the model building process, and what previous studies have used similar models (model testing approach).

See the following: A three-player game model for promoting the diffusion of green technology in manufacturing enterprises from the perspective of supply and demand.https://doi.org/10.3390/math8091585

A stochastic differential game of low carbon technology sharing in collaborative innovation system of superior enterprises and inferior enterprises under uncertain environment, https://doi.org/10.1515/math-2018-0056

Information fusion for future COVID-19 prevention: continuous mechanism of big data intelligent innovation for the emergency management of a public epidemic outbreak.

Answer: Thanks to the reviewer's comments, according to the reviewer's comments and with reference to related studies, we have revised the title of the fourth and fifth parts of the article in conjunction with the content of the study, and added the process of modeling and the related study of applying the model at the beginning of the fourth part. The specific modifications are shown in the markup of the fourth and fifth parts of the revised manuscript.

Q12: Please consider this structure for manuscript final part. -Discussion-Conclusion-Managerial Implication-Practical/Social Implications

Answer：Thanks to the reviewer's comments, we have reorganized the article according to the structure suggested by the reviewers, specific textual changes are reflected in the markup of the revised manuscript.

Q13: Please make sure your conclusions' section underscores the scientific value-added of your paper, and/or the applicability of your findings/results. Highlight the novelty of your study. In addition to summarizing the actions taken and results, please strengthen the explanation of their significance. It is recommended to use quantitative reasoning comparing with appropriate benchmarks, especially those stemming from previous work.

Answer：Thanks to the reviewer's comments, according to the reviewer's comments, we further emphasize the scientific value and novelty at the beginning of the conclusion. Meanwhile, in the new dicussion section, we take government subsidies and corporate pricing as a point of discussion to explore their further value in the light of previous studies, and the specific textual changes can be found in the markups in Part 5 and 6 of the revised manuscript.

The article needs additional financial support as follows: Specific grant numbers: Project of Humanities and Social Sciences Foundation of the Ministry of Education (Specific grant numbers: 18XJAZH004); Initials of authors who received each award: H H; Full names of commercial companies that funded the study or authors: Ministry of Education of China; Initials of authors who received salary or other funding from commercial companies: W Z; URLs to sponsors’ websites: https://www.sinoss.net/. The funders had no role in study design, data collection and analysis, decision to publish, or preparation of the manuscript.

---

## [Decision Letter · Decision Letter 2]

21 Nov 2023

Network evolution of diffusion in enterprise digitalization and intellectualization transformation: A technology–organization–environment framework perspective

PONE-D-23-06664R2

Dear Dr. HU,

We’re pleased to inform you that your manuscript has been judged scientifically suitable for publication and will be formally accepted for publication once it meets all outstanding technical requirements.

Best regards,

Prof. (Assist.) Donato Morea, Ph.D.

Academic Editor

PLOS ONE

Reviewers' comments:

Reviewer's Responses to Questions

**Comments to the Author**

1. If the authors have adequately addressed your comments raised in a previous round of review and you feel that this manuscript is now acceptable for publication, you may indicate that here to bypass the “Comments to the Author” section, enter your conflict of interest statement in the “Confidential to Editor” section, and submit your "Accept" recommendation.

Reviewer #1: (No Response)

2. Is the manuscript technically sound, and do the data support the conclusions?

Reviewer #1: (No Response)

3. Has the statistical analysis been performed appropriately and rigorously? 

Reviewer #1: (No Response)

4. Have the authors made all data underlying the findings in their manuscript fully available?

Reviewer #1: (No Response)

5. Is the manuscript presented in an intelligible fashion and written in standard English?

Reviewer #1: (No Response)

6. Review Comments to the Author

Reviewer #1: The manuscript has significantly improved as compared to the previous version. Indeed, the authors tried to improve it, and the main weaknesses are solved.

Thus, in my opinion, the manuscript is recommendable for publication.

7. PLOS authors have the option to publish the peer review history of their article (what does this mean?). If published, this will include your full peer review and any attached files.

Reviewer #1: No

---

## [Editor Report · Acceptance letter]

11 Dec 2023

PONE-D-23-06664R2 

Network evolution of diffusion in enterprise digitalization and intellectualization transformation: A technology–organization–environment framework perspective 

Dear Dr. HU:

I'm pleased to inform you that your manuscript has been deemed suitable for publication in PLOS ONE. Congratulations! Your manuscript is now with our production department. 

Kind regards, 

on behalf of

Professor (Assistant) Donato Morea 

Academic Editor

PLOS ONE